# Selective Modulation of Hippocampal Theta Oscillations in Response to Morphine versus Natural Reward

**DOI:** 10.3390/brainsci13020322

**Published:** 2023-02-14

**Authors:** Shole Jamali, Mohsen Parto Dezfouli, AmirAli Kalbasi, Mohammad Reza Daliri, Abbas Haghparast

**Affiliations:** 1Neuroscience Research Center, School of Medicine, Shahid Beheshti University of Medical Sciences, Tehran P.O. Box 19615-1178, Iran; 2School of Cognitive Sciences, Institute for Research in Fundamental Sciences, Tehran P.O. Box 19395-5531, Iran; 3Department of Mechatronics, Faculty of Electrical Engineering, K. N. Toosi University of Technology, Tehran P.O. Box 16315-1355, Iran; 4Biomedical Engineering Department, School of Electrical Engineering, Iran University of Science and Technology, Tehran P.O. Box 16846-13114, Iran

**Keywords:** reward, food, morphine, hippocampal CA1 region, local field potential, theta oscillations

## Abstract

Despite the overlapping neural circuits underlying natural and drug rewards, several studies have suggested different behavioral and neurochemical mechanisms in response to drug vs. natural rewards. The strong link between hippocampal theta oscillations (4–12 Hz) and reward-associated learning and memory has raised the hypothesis that this rhythm in hippocampal CA1 might be differently modulated by drug- and natural-conditioned place preference (CPP). Time–frequency analysis of recorded local field potentials (LFPs) from the CA1 of freely moving male rats previously exposed to a natural (in this case, food), drug (in this case, morphine), or saline (control) reward cue in the CPP paradigm showed that the hippocampal CA1 theta activity represents a different pattern for entrance to the rewarded compared to unrewarded compartment during the post-test session of morphine- and natural-CPP. Comparing LFP activity in the CA1 between the saline and morphine/natural groups showed that the maximum theta power occurred before entering the unrewarded compartment and after the entrance to the rewarded compartment in morphine and natural groups, respectively. In conclusion, our findings suggest that drug and natural rewards could differently affect the theta dynamic in the hippocampal CA1 region during reward-associated learning and contextual cueing in the CPP paradigm.

## 1. Introduction

Besides drug abuse, addiction to natural rewards such as food is also a major public health concern. In addition to drug abuse such as that of morphine, cocaine, and methamphetamine, addiction can also refer to compulsive behaviors associated with natural rewards such as food, gambling, sex, and shopping [1,2,3,4,5].

The drug and non-drug cues target the reward system [6,7]. Natural cues such as food, especially palatable food, affect the reward circuits, which leads to food reinforcement and, consequently, obesity [8,9]. There has been considerable interest in the common neurobiological mechanisms underlying drug and natural rewards in recent years [10,11,12,13], mainly due to significant health concerns associated with prevalent disorders such as addiction and obesity [14,15]. Previous studies have focused more on behavioral and neurochemical similarities and differences between drug rewards and natural rewards than on electrophysiological characteristics of the reward system [10,16,17].

Furthermore, drug dependency treatment influences responses to natural rewards since abused drugs and natural cues seem to utilize the same reward circuit [1], which poses a problem for the treatment of addiction since natural rewards such as food and sexuality play a large part in people’s lifestyles. A disruption of the natural reward system may result in anxiety, depression, and other reward-related disorders. As a result, the important question is, “Are there any differences in the neuronal pathways mediating the action of natural rewards and those associated with drugs?”.

Drug- and food-induced rewards are mediated by the brain reward system, which includes the ventral tegmental area (VTA), nucleus accumbens (NAc), hippocampus (HIP), medial pre-frontal cortex (mPFC), and amygdala [18,19,20,21,22,23]. The HIP plays a crucial role in various spatial and contextual learning and memory functions [24,25], including expression, consolidation, and retrieval of drug and food rewards [26]. HIP receives substantial dopaminergic input from the VTA [27] and sends glutamatergic projections to the NAc, which is thought to act as an integration site between spatial/contextual information from the dorsal hippocampus [26] and emotional information from the basolateral amygdala [28] and locus coeruleus [29].

The hippocampal CA1 region is known for its role in the cognitive representation of specific locations in space [30], which has an important role in CPP acquisition and expression [31]. Recent findings indicate that HIP/NAc coupling increases after cocaine-CPP by strengthening of hippocampal CA1 inputs to NAc during conditioning [32]. These studies suggest that the CA1 region is an essential component of a neural circuit that mediates the formation of reward-associated representations. Despite this, direct evidence of the comparative role of CA1 neural activity in morphine and food rewards is absent in freely moving rats during conditioning preference tasks.

Many cognitive operations require dynamic coordination of activity across distributed groups of neurons. Oscillations in neural population activity are important for temporal coordination of neural activity on a relatively fast time scale [33]. Among brain rhythms, hippocampal theta oscillations (4–12 Hz) are prominent during active behaviors and are associated with mnemonic processing related to spatial exploration, navigation, and reward-related behaviors [34].

The evidence that CA1 plays a significant role in reward-related behavior, as well as its strong connections with learning, attention, and memory [35], and the theta rhythm’s role in reward-related behaviors led us to conclude that drug and natural rewards may modulate theta rhythm differently in the hippocampal CA1 region. The hypothesis was tested by using morphine as a drug in comparison with palatable food (biscuit) as a natural reward.

A number of studies have investigated the biochemical, cellular, and molecular mechanisms of natural- or drug-related responses. However, few comparative studies have explored the differences in neural activity between natural rewards and drug rewards. To our knowledge, the present study is the first report that compares hippocampal CA1 neural activity between natural (food) and drug (morphine) rewards in an equivalent, parallel, behavioral condition. During the performance of a classical CPP task, local field potentials (LFPs) were recorded from the hippocampal CA1 of freely moving rats. Note that the animals had previously been exposed to natural (in this case food), drug (in this case morphine), or saline (as a control) reward cues. We hypothesized that theta oscillations, as the dominant hippocampus activity, might play a role in discriminating these types of rewards in terms of neuronal responses.

## 2. Materials and Methods

### 2.1. Animals and Surgery

Thirty-six male Wistar rats (Pasteur Institute, Tehran, Iran) weighing 220–270 g at the start of each experiment were maintained in controlled conditions of light (12/12 h light/dark cycle), temperature (25 ± 2 °C), and humidity (55 ± 10%). Rats were food restricted to 80–85% of their free-feeding body weight before the conditioning phase, and following that, they had free access to water (two rats per cage) while experiencing a feeding regimen to maintaining the body weight during the experiment. All procedures were approved by the Ethics Committee of Shahid Beheshti University of Medical Sciences (IR.SBMU.SM.REC.1395.373), Tehran, Iran, and were in accordance with the National Institutes of Health Guide for the Care and Use of Laboratory Animals (NIH Publication, 8th edition, revised 2011). The rats were anesthetized intraperitoneally with a mixture of ketamine and xylazine (100/10 mg/kg) [36] and placed in a stereotaxic apparatus (SR-8N, Narishige, Japan). Rats were implanted with bipolar recording electrodes recording local field potentials (LFPs) in the CA1 at the following coordinates: anteroposterior (AP): −3.5 mm from bregma, lateral (L): ±2.6 mm, dorsal–ventral (DV): −2.6 mm. The reference and ground screws were inserted in the skull. The rats were allowed to recover for ten days following surgery.

### 2.2. Drugs

Ketamine and xylazine were obtained from Alfasan Chemical Co., Woerden, Holland. Morphine sulfate (Temad, Iran) was dissolved in physiological saline (0.9% NaCl) and administered by a subcutaneous (s.c.) route at the dose of 5 mg/kg in the conditioning phase.

### 2.3. Conditioned Place Preference Paradigm

All rats experienced an unbiased, counterbalanced conditioned place preference procedure [37]. LFP recording data were acquired from the hippocampal CA1 of rats during freely moving behavior in the pre- and post-test of CPP. The CPP procedure involves three phases: pre-conditioning, conditioning, and post-conditioning. The experiments were performed in a three-compartment Plexiglas CPP apparatus, which consisted of two equal-sized compartments as the main chambers for conditioning reward, and a smaller chamber (null) connecting the two main chambers. The floor texture (smooth or rough) and wall stripe pattern made the two main compartments different. One of the compartments’ walls were striped horizontally, and the other compartment had a vertically striped wall.

The behavior was monitored through a 3CCD camera (Panasonic, Japan) positioned above the apparatus. Data were analyzed by Ethovision software (Noldus Information Technology, The Netherlands), a video tracking system for automation of behavioral experiments that was programmed to simultaneously trigger the onset of behavioral tracking and the beginning of LFP recording. Therefore, behavior and electrophysiological sessions were recorded in a synced manner.

During the pre- and post-test, rats freely explored the entire arena for 10 min while they were connected to the LFP recording cable. Rats that showed an inherent preference >70% for either of the main compartments of the CPP were removed from the experiment (three rats in total) [38]. The distance traveled and time spent in each of the compartments were recorded. The CPP scores were calculated by subtracting the time spent in the unrewarded paired compartment (unrewarded compartment) from that spent in the rewarded paired compartment (rewarded compartment). The total distance traveled (in cm) was considered the index of locomotor activity for each animal.

#### 2.3.1. Conditioning Phase (Saline, Morphine, Food)

On the first day of the conditioning phase, each animal received morphine (5 mg/kg, s.c.) in the morning and was confined to one chamber for 45 min; about 6 h later, animals were injected with saline, as the vehicle (1 mL/kg, s.c.), and were confined to the other main chamber of the CPP compartment for 45 min. On alternate days, morphine and saline injections were arranged in a counterbalanced manner. The third day of conditioning was the same as the first day. During this phase, access to other chambers of the CPP box was blocked. In the natural (food) group, on the first day of the conditioning period, in the morning session, food-restricted animals received biscuit (6 g) as a reward in the middle of one main compartment, and 6 h later, they were placed into the other compartment with no food; each session lasted 45 min. On the following days, biscuit and no-food sessions were arranged in a counterbalanced manner over the conditioning period. Throughout the experiment, animals were maintained on a restricted diet at 80–85% of their free-feeding weight but had access to water ad libitum [38,39]. As a control group in the saline group, animals just received saline in either of the main compartments (Figure 1A). Each animal was injected with saline (1 mg/kg, s.c.) in the morning before being placed in one of the main CPP chambers for 45 min. Six hours later, they were again injected with saline and confined to the other CPP chamber for 45 min. The conditioning phase lasted for three days and consisted of two 45 min sessions per day. The number of animals in each saline, morphine, and food group was 12 rats.

#### 2.3.2. Post-Conditioning Phase (Post-Test)

Twenty-four hours after the conditioning phase, animals were tested for 10 min (post-test trial) in which they could explore the entire CPP arena; this was similar to the pre-test session (Figure 1A). The behavioral and LFP data were measured during this session. The CPP scores were calculated as the time spent in the rewarded compartment minus the time spent in the unrewarded compartment. The total distance traveled (cm) was considered as the locomotor activity index for each animal. At the end of the experiments, the electrode trace was marked with the electrical lesion (25 μA, 10 s) before the animals were perfused by isotonic saline followed by 10% formalin. Brains were sliced (150 μm) using a vibrating microtome (Campden Instruments, Traunstein, Germany). Electrode tip traces were localized using a light microscope and were confirmed using a rat brain atlas [40] (Figure 1B). Only the animals with confirmed electrode placements were included in the data analysis (one rat was excluded from the study).

### 2.4. Behavioral and Electrophysiological Recordings

Behavior was recorded with a digital video camera (30 frames per second), and each rat’s movements were tracked by an automated system that synchronized behavioral data and electrophysiological recordings. The spatial position was defined as the center of the animal body in each frame. During the experiments, a lightweight and flexible cable was connected to the pins on the head-stage pre-amplifier. Recordings, digitalization, and filtering of neural activities were performed using a commercial acquisition processor (Niktek, Tehran, Iran).

### 2.5. Data Analysis

All analyses were conducted using Matlab 2016b (The MathWorks, Inc., Natick, MA, USA). The raw LFP signals were band-pass filtered at 0.01–90 Hz and sampled at 1000 Hz. The saturated data were removed manually based on abnormal signals. Next, the data’s baseline was removed using the empirical mode decomposition approach [41]. Finally, power spectrum trials with power three times more than the mean ± SEM of the average were excluded. The number of involved animals in data analysis in the saline, morphine, and natural groups was 10, 12, and 10, respectively.

After pre-processing, the remaining trials were considered for the main analyses. We defined trials based on every single entrance from null to each main compartment (null–rewarded and null–unrewarded). For each trial, the time scale from 1 s before to 1 s after the entrance was considered as the trial time. We compared the LFP trials between two null–rewarded and null–unrewarded sessions during the post-test period. Time analyses were performed based on the grand mean of LFPs within each condition by averaging LFP signals across trials of that condition. In the frequency-domain, the power spectral was calculated based on the Welch approach using the custom-built Matlab code “pwelch”. Similarly, band power was extracted using custom-built Matlab code “bandpower”.

The time–frequency map was estimated using continuous wavelet transform (CWT) based on Morlet mother wavelet. To this end, several wavelets in a different time and frequency scales were convolved to the LFP signals to determine the power of those time–frequency scales [42].

In each trial, the analyses were computed for one second before and one second after the entrance to each main compartment from null, which were considered null–rewarded and null–unrewarded trials.

### 2.6. Statistics

Data were expressed as mean ± SEM, and their normality was tested using the Kolmogorov–Smirnov test. The data were processed by Matlab 2016b. The paired *t*-test was used to compare the time spent in the rewarded and unrewarded compartments and compare the CPP score between pre-test and post-test in each group. One-way ANOVA followed by a post hoc Newman–Keuls test was used to compare the CPP scores, or the distance traveled in experimental and control groups. Statistical comparisons between two conditions of the same animals and between two groups of animals were performed using paired and unpaired *t*-tests, respectively. Results were considered statistically significant when *p* < 0.05.

## 3. Results

The LFP data from the hippocampal CA1 area in freely moving rats were recorded while animals performed a CPP experiment (Figure 1, see Section 2 Materials and Methods). Reward-induced effects were investigated using behavioral and electrophysiological data between rewarded and unrewarded conditions for three groups of animals that received saline, morphine, or natural food during the CPP task.

### 3.1. Comparing Morphine- and Food-Induced Conditioned Place Preference

In the first step, the time spent in each compartment was considered as a behavioral parameter to measure an animal’s preference for each compartment. A comparison of this parameter between the rewarded compartment vs. unrewarded compartment in each group indicated rats’ behavioral tendency due to that kind of condition. Behavioral results show no difference between the two conditions (rewarded vs. unrewarded) in the saline group (*t*-test, *p* = 0.23; *n* = 10, Figure 2A, left panel) in which animals received a similar amount of saline in both compartments (Figure 1). As expected, a significant increase was observed in the time spent in the rewarded compartment in comparison to the unrewarded compartment in the morphine group during the post-test session (*t*-test, *p* < 0.05; *n* = 12, Figure 2A, middle panel), as animals received morphine in the rewarded compartment and saline in the unrewarded compartment (Figure 1). There is no significant difference between the time spent between these compartments in the pre-test session (*t*-test, *p* = 0.74; Figure 2A, middle panel). The results showed that exposure to the palatable food (biscuit) increases the time spent in the rewarded compartment in comparison to the time spent in the unrewarded compartment (*t*-test, *p* < 0.05; *n* = 10, Figure 2A, right panel) in the post-test session, while there is no significant difference between the time spent in the rewarded and unrewarded compartments during pre-test (*t*-test, *p* = 0.11; Figure 2A, right panel).

Moreover, we compared the reward-induced behavioral modulation between pre- and post-test, considering the CPP score. As a result, no significant difference is observed between the pre-test and post-test CPP scores in the saline group (*t*-test, *p* = 0.21; Figure 2B, left panel). The CPP score in the post-test was increased significantly compared to that in the pre-test in the morphine-treated group (*t*-test, *p* < 0.01; Figure 2B, middle panel). Therefore, saline did not induce CPP. These results show that rats preferred the morphine-paired (5 mg/kg, s.c.) compartment to the saline-paired compartment. Moreover, the paired *t*-test indicates that biscuit (as a food reward) increased the CPP score of the post-test compared to the pre-test (*t*-test, *p* < 0.01; Figure 2B, right panel). As illustrated in Appendix A, one-way ANOVA followed by the Newman–Keuls test (F (2, 31) = 0.078, *p* = 0.64) showed that the CPP score was increased in morphine and natural groups compared with the saline group (*p* < 0.05), while there is no significant change between morphine and natural groups. This finding indicates that drugs and FD stress did not affect motor activity.

Based on the paired *t*-test (*p* > 0.05), there is no significant difference between the distance traveled in the pre-test and post-test in each group (saline, morphine, and natural) (Figure 2C). One-way ANOVA followed by Newman–Keuls multiple comparison test (F (2, 31) = 0.0265; *p* = 0.466; Appendix A) showed no significant difference in distance traveled among the saline, morphine, and food groups during the post-test phase. This finding suggests these treatments did not affect the locomotor activity.

Results showed that time spent by the animals in the morphine (*p* < 0.01) and natural (*p* < 0.001) groups was significantly more than time spent by the animals in the saline group, while there is no significant difference between morphine and natural groups (F (2, 31) = 0.094, *p* = 0.24) (Appendix A).

### 3.2. Comparing Local Field Potential Activity for Morphine and Food Reward in Conditioned Place Preference Paradigm

A single entrance from null to each main compartment (null–rewarded and null–unrewarded) is defined in one trial lasting from 1 s before to 1 s after the entrance. The analyses were performed on all trials’ LFP data for the null–rewarded and null–unrewarded sessions during the post-test, separately. This analysis allowed us to compare oscillatory activity (pattern) exhibited during the behavioral approach to the reward-paired compartment and unrewarded compartment.

We compared these two rewarded and unrewarded conditions for the three groups of reward (saline, morphine, and natural food), separately (Figure 3, Figure 4 and Figure 5).

Figure 3A, Figure 4A and Figure 5A show the pattern of movement among the compartments for a sample rat. The time–frequency analysis was performed on the two defined trial groups separately (rewarded and unrewarded). To this end, we used a continuous wavelet transform (CWT; see Section 2 Materials and Methods) (Figure 3B, Figure 4B and Figure 5B). Finally, the power spectrum between the rewarded vs. unrewarded trials was calculated for each 1000 ms null and 1000 ms rewarded/unrewarded LFP signal, separately (Figure 3C, Figure 4C, and Figure 5C).

#### 3.2.1. Hippocampal CA1 Theta Activity Relation to Approach to Rewarded and Unrewarded Compartments during the Post-Test in Saline-Treated Animals

The behavioral data showed that saline (1 mg/kg, s.c.) did not induce CPP. The red traces of freely behaving animals also represent the same result (Figure 3A). We examined the hippocampal CA1 LFPs during the post-test in the two main compartments. As expected, time–frequency (Figure 3B) and power spectrum (Figure 3C) results showed that there was no significant difference in theta band regarding the chamber entries (paired *t*-test, *p* > 0.05). In other words, no significant change was detected in the theta band during null–rewarded and null–unrewarded sessions. Therefore, these results lead us to report place difference between the two main compartments as the only variable that could not change theta activity when animals entered each compartment.

#### 3.2.2. Hippocampal CA1 Theta Activity Relation to Approach to Rewarded and Unrewarded Compartments during the Post-Test with Morphine Reward

The behavioral data showed that the morphine-treated animals preferred the drug compartment (rewarded compartment) to the saline compartment (unrewarded compartment) (Figure 2A,B; middle). Red traces (Figure 4A) show that the rats spent more time in the morphine (reward)-paired side. The time–frequency analysis of hippocampal CA1 LFPs for null–rewarded and null–unrewarded sessions showed that the theta peaked in the time window −727 to −153 before entrance to the unrewarded compartment (FDR-corrected *t*-test, *p* < 0.05; Figure 4B). The LFP mean power distribution shows an increase in the mean theta power before the entrance to the unrewarded compartment, and this effect was decreased following the entrance to the unrewarded compartment from null (null–unrewarded) (paired *t*-test, *p* < 0.01; Figure 4C). These results indicate that the theta band oscillation with morphine treatment acts differently when the animals enter the rewarded (morphine) compartment compared to the unrewarded (saline) compartment.

#### 3.2.3. Hippocampal CA1 Theta Activity Relation to Approach to Rewarded and Unrewarded Compartments during the Post-Test with Food Reward

Animals that received biscuits as a natural reward preferred the rewarded (food) compartment rather than the unrewarded compartment (Figure 2A,B; right). Figure 5A indicates the preference for the rewarded compartment rather than the opposite compartment; red traces show the rats spent more time in the rewarded (food) side. We examined the hippocampal CA1 LFPs of rats during the post-test phase for both null–unrewarded and null–rewarded trials. The time–frequency analysis shows that the power in the theta band peaked after the rat entered the rewarded compartment from 136–420 ms (FDR-corrected *t*-test, *p* < 0.05; Figure 5B)**.** The mean power distribution shows that the mean power in the theta band was significantly higher in the rewarded (food) compartment than the unrewarded compartment (paired *t*-test, *p* < 0.05; Figure 5C). These results show that when the animal entered the side in which it received a biscuit (palatable food as a natural reward), the hippocampal CA1 theta oscillation acted differently compared to when the animal entered the opposite compartment (unrewarded compartment).

#### 3.2.4. Comparing Hippocampal CA1 Theta Pattern between Morphine- and Natural-Induced CPP

Toward comparing the theta band activity of the hippocampal CA1 area of rats that received morphine as a drug reward with those that received biscuit as a natural reward during conditioning days, we measured the difference in CA1 LFP activity by subtracting saline as a control group from the morphine (Figure 6A) and food (natural) groups (Figure 6B). The time–frequency analysis indicates that in the morphine group, the theta mean power of CA1 was increased before the entrance to the unrewarded compartment (Figure 6A) while in the natural group, theta power increased after the entrance to the rewarded compartment (Figure 6B). In line with the above results, the time–frequency representation based on CWT of the difference between morphine and natural groups in the one-second time window before and after the entrance to rewarded and unrewarded compartment (Figure 6C) confirms the increase in the theta power before the entrance to the unrewarded compartment (as shown in red) and after the entrance to the rewarded compartment (as shown in blue). One-way ANOVA of the mean power modulation (unrewarded minus rewarded) indicates a significantly higher modulation in the theta power of the morphine group compared to the saline or natural group (F (2, 31) = 11.7, *p* < 0.001)) before the entrance to the unrewarded compartment (Figure 6D, top). No significant modulation was observed between these groups in other frequency bands.

Similar analyses for the after entrance period exhibited a higher theta power of the natural group than the saline and morphine groups (F (2, 31) = 3.39, *p* < 0.05; Figure 6D, bottom). No significant difference was observed between these three groups in other frequency bands.

## 4. Discussion

We examined the theta activity of hippocampal CA1 in CPP induced by natural (food, in this case) and drug (morphine, in this case) rewards. Based on our findings, in the animals that received morphine as a drug reward, the approach to the unrewarded compartment was accompanied by enhanced theta activity (theta power) in hippocampal CA1 LFPs, compared with the rewarded compartment approach. In those animals that received food as a natural reward, the CA1 neural dynamic was modulated during the rewarded side approach. The theta power was increased when animals entered the rewarded compartment. The above results point to the conclusion that the hippocampal CA1 theta activity had a different pattern between the null–rewarded and the null–unrewarded sessions during the post-test session of both drug- and natural-CPP. Meanwhile, in the saline group, there is no significant difference between approaches to the rewarded and unrewarded compartments. As mentioned earlier, the rewarded and unrewarded compartments are not only different in terms of the presence or absence of rewards, but also the particular spatial and textural cues defining each compartment. Therefore, the difference between theta activity in the null–rewarded and null–unrewarded sessions is due to the effect of morphine or food as a reward.

We also show that drug reward (morphine in this case) and natural reward (food in this case) could differently affect the reward-associated hippocampal CA1 dynamic in the CPP paradigm. The subtracting of the CA1 LFP activity of the saline group from the morphine group or natural group showed that theta’s maximum power occurred before entering the unrewarded compartment and after the entrance to the rewarded compartment in the morphine and natural groups, respectively.

Studies showed that the hippocampal CA1 is a crucial area for place coding, which plays an important role in reward-related memory [43]. However, it remains unclear which sources provide the motivational information for the hippocampus during ongoing reward-related behaviors. The CA1 receives the most dopaminergic innervation from the VTA [44]. Additionally, the hippocampus receives dopamine innervation from the LC. This dopamine signal may be an important reward predictor that could arrive in the hippocampus or other important reward-related areas such as the NAc and amygdala during reward-associated behavior [45]. The other essential source, the mPFC, in which the neural activity is correlated with different aspects of seeking and planning of reward-associated behavior, including choice location (place preference), could influence the activity of several brain regions including the HIP, NAc, and VTA [46]. Furthermore, striatal neurons play an important role in coding reward by cooperating with dopamine signals. Studies show that the strength of glutamate excitatory inputs from the hippocampus to the accumbens is associated with reward-related behaviors [21].

An animal’s ongoing behavior affects the hippocampal theta activity, and the activity of this oscillation represents the line state of the hippocampus [47]. Hippocampal theta oscillations depend on pyramidal cells and the inhibitory network of GABAergic interneurons [48]. The hippocampal CA1 area contains two different neural populations: place and reward cell populations. The place cells encode the animal’s location while the reward cells encode the reward location. The reward cells may project to the NAc as a specific target [49]. This point is consistent with observations in the hippocampal CA1, in which neurons projecting to the NAc are more likely to be active near reward locations than those that project to other areas [50].

On the other hand, the hippocampal interneuron network plays an important role in the timing of pyramidal cells in the theta cycle via GABAA and GABAB receptors [49]. Lansink et al. showed that theta’s power was increased during a cued approach compared to a non-cued approach regarding white cued chamber animals that received sucrose as a reward [51]. It is consistent with this finding that in animals that received biscuit as a natural reward, theta power increased in the rewarded chamber compared to the unrewarded chamber. Both studies show that reward expectancy increases hippocampal theta power. Sjulson et al. found that the hippocampal/nucleus accumbens activity increased after cocaine-CPP expression because of strengthened place cell activity during conditioning [32] while German et al. showed that morphine-CPP resulted in less firing of accumbal neurons in the morphine-paired location, suggesting that the association of drug reward and spatial location may occur through different mechanisms [52]. Therefore, it is possible that morphine conditioning leads to a lower firing rate of hippocampal CA1 neurons during the approach to the morphine-paired compartment than saline compartment (unrewarded compartment).

The present study showed that, in the morphine group, the theta power increased before the entrance to the unrewarded chamber, while in the natural group, theta power increased after entering the rewarded chamber. The issue is how each of these two different rewards, morphine and food, affects the activity of hippocampal cells, neurotransmitters, and receptors involved in the reward circuit, and consequently, the theta activity. There are very few studies comparing natural and drug rewards, and there is no comparative study comparing the mechanisms underlying morphine reward (as a drug reward) and food reward (as a natural reward). Studies indicate that morphine pairing in the CPP apparatus decreased the number of thin dendritic spines in the hippocampus. This effect was not observed when the animal received morphine in the home cage or when the animal was trained via an unpaired morphine-CPP [53].

In contrast, other studies show that operant training with palatable food results in a significant increase in spine density in the NAc, mPFC, and OFC, important areas for hedonic aspects of food [54]. Therefore, a possible explanation is that food and morphine may have a different effect on CA1 neuronal plasticity and other crucial reward-related areas connected to CA1. This differential dendritic spine plasticity induced by morphine or food may differently affect the theta activity of CA1.

Hippocampal theta activity is influenced by neurotransmitters and neuromodulators, including dopamine, glutamate, acetylcholine, and GABA, whose their signaling is affected by drug and natural rewards. The mesolimbic dopamine pathway is the initial site for drugs while the role of this pathway in food intake is more nuanced. Results show that the animals could respond to the hedonic aspects of food in the absence of dopamine [55]. Other studies have shown that dopamine depletion reduces dopamine signaling with food-related reward activity [56]. Food intake induces DA release in the striatum, associated with the rewarding properties of food, and mesolimbic DA is crucially involved in the motivation to obtain food [57]. Results also show that hippocampal dopamine receptors are involved in the acquisition and expression of morphine-CPP in rats [31]. Therefore, dopamine is critically involved in drug and natural rewards. Enhanced extracellular dopamine decreases the frequency of hippocampal theta oscillations modulated with secondary alterations in the serotonergic neuromodulatory system [58]. Intraventricular infusion of DA reuptake blocker produced an increase in the firing rate and modulation of the medial septal neurons, consequently increasing the power of the hippocampal theta rhythm [59]. In another study, direct microinjection of DA receptor agonists into the septum or dorsal hippocampus increased the release of Ach that plays a crucial role in the generation and modulation of theta activity [60]. Generally, the activation of GLU receptors or a decrease in GABAergic tonus in the VTA leads to enhancement of DA release into the hippocampus that could affect the theta frequency and power [59]. Taken together, the different CA1 theta patterns between the natural (food) and drug (morphine) rewards may depend on their specific effects on the dopamine signaling.

Besides dopamine signaling, glutamatergic, cholinergic, and GABAergic systems could modulate the hippocampal theta activity [61]. The hippocampal theta oscillation is regulated by glutamate activation of pyramidal and granule cells via NMDA and AMAP receptors [62,63]. Glutamate receptors in reward-associated areas such as the VTA, NAc, mPFC, and hippocampus are components of the mechanisms underlying reward and modulate the firing pattern of dopaminergic neurons in the reward system. Studies showed that the AMAP and NMDA receptor antagonists abolish the theta activity in the hippocampus. In addition, the combination of NMDA receptor blockers and atropine or scopolamine eliminates all hippocampal theta activity.

On the other hand, NMDA and AMPA receptor antagonists block morphine-CPP [64]. Hippocampal NMDA lesions did not impair the performance in linear track tasks to obtain food reward while learning of a continuous spatial alternation task to obtain food was impaired [65]. The activity of glutamate signaling in the reward circuit may be affected differently by different reward inducers such as food or morphine and it also depends on the behavioral task to obtain a reward.

Studies show that lesions of the medial septum and diagonal band of Broca (MS-DBB) abolish theta oscillations in the entorhinal cortex [66]. Muscarinic and nicotinic receptors are also involved in theta regulation. Hippocampal interneurons and their rhythmic discharge are the exclusive targets of the cholinergic and GABAergic septo-hippocampal projection [61]. Therefore, the activation of septo-hippocampal cholinergic terminals present in all hippocampal layers could generate and regulate the hippocampal theta activity that may be affected differently by food or morphine treatment during conditioning to these different reward inducers. Intra-CA1 administration of anticholinesterase and muscarinic receptor antagonist (atropine) significantly potentiates and inhibits morphine-induced CPP, respectively [67].

Furthermore, bilateral injections of nicotinic receptor antagonist into the CA1 significantly inhibits the morphine-CPP. It may be concluded that the muscarinic and nicotinic receptors of the hippocampal CA1 regions play an important role in morphine reward [67,68]. Pre-treatment with cholinergic antagonists could block drug- or food-reinforced responding. The muscarinic agonist modifies food- and cocaine-reinforced behavior [69]. This evidence leads to the idea that cholinergic activity as a crucial component of the reward system may change differently depending on the reward type.

GABA signaling is also involved in hippocampal theta activity. Basket and chandelier cells induce the perisomatic inhibition of pyramidal cells via GABAA receptor-mediated IPSPs. GABAB receptor blockade may enhance cognitive task performance by activating hippocampal theta and gamma rhythms in freely behaving rats [70]. Intra-CA1 administration of baclofen and phaclofen could decrease and increase the acquisition of morphine-CPP, respectively. It is concluded that the GABA (B) receptors in the dorsal hippocampus may play an active role in morphine reward [71]. Intra-CA1 administration of the GABAA receptor agonist muscimol and GABA (A) receptor antagonist bicuculline significantly inhibited and elicited morphine-CPP [72]. Our data indicated that the GABA (A) receptors of the hippocampal CA1 regions might play an important role in the acquisition and expression of morphine-induced place preference.

It has been demonstrated that ICV injections of m-, d-, and k-opioid agonists, or microinjections of these agonists into the NAc shell, prevent morphine-induced CPP in a dose-dependent manner [73]. Kisspeptin-10–kissorphin (KSO) inhibits acquisition and expression of morphine-CPP by its antiopioid activity [74]. A study indicates that μ-opioid receptor is associated with response to food reward in humans [75]. Opioid agonists enhance food intake and hedonic responses to palatable foods, while opioid antagonists decrease them [76]. It would be interesting to design a comparative study to examine and compare the role of opioid receptors in response to morphine and food rewards.

Increasing endogenous glucagon-like peptide-1 (GLP-1) levels in the CNS could attenuate the rewarding effects of both morphine [77] and food [78], likely by activating pre-synaptic GLP-1 receptors on glutamatergic terminals, which facilitate synaptic excitation of dopamine neurons in the VTA.

Neuropeptides such as galanin and NPY regulate drug and food intake in opposite directions. Both neuropeptides increase food intake, but galanin decreases, whereas NPY increases, cocaine reward [55]. Therefore, it may be possible that the differential hippocampal theta activity observed between morphine and food rewards in the current study is due to a differential effect of these two types of reward on the activity of neurotransmitter systems.

In conclusion, the present study shows that hippocampal theta activity was affected differently by morphine- and food-CPP. The theta pattern is also different between the rewarded and unrewarded states in CPP in both food and morphine groups. It seems that despite the overlapping neural circuit in natural reward and drug reward, the neuronal activity in the hippocampus, as an important area in the reward circuit, could be affected differently by morphine (as a drug) and food (as a natural reward). Very few comparative studies have focused on the difference between the neural mechanisms underlying natural and drug rewards using a parallel behavioral condition. The current study compared morphine and food rewards in two separate animal groups but not in the same group. It would be more convincing to compare morphine- and food-CPP in the same animal. In order to compare drug to natural rewards within the same animal, we need a model in which the animal self-administers both rewards (drug vs. natural) in alternate sessions, and reward-specific seeking is triggered by specific reward-associated cues [12]. Therefore, one important goal of future research would be investigating the complexity of neuronal mechanisms underlying natural and drug rewards that are the basis for pathological corruption of these pathways, leading to obesity and addiction, by considering the problem of genetically regulated individual differences in sensitivity to drugs of abuse and food reward [79]. Orsini et al. showed that two different strains of mice (DBA/2J (DBA) and C57BL/6J (C57)) differ in sensitivity to morphine- and cocaine-CPP [79,80]. Not only strain but also sex affect response to reward cues in mice. These studies highlight the importance of discovering the genetic mechanisms underlying reward- and addiction-related behaviors [81].

Finally, one interesting speculation derived from the present study is that neuronal activity within or between reward-associated areas such as the VTA, NAc, AMYG, and mPFC may specifically change with drug and natural rewards. Using multiarray recording is suggested to improve the quality of data pull. Optogenetic inhibition of hippocampal CA1 inputs from reward-associated regions could reveal which circuit is more involved in morphine (as a drug) and food (as a natural reward) rewards. Our future focus is the connectivity between the hippocampal CA1 and NAc in natural- and morphine-induced rewards.

## Figures and Tables

**Figure 1 brainsci-13-00322-f001:**
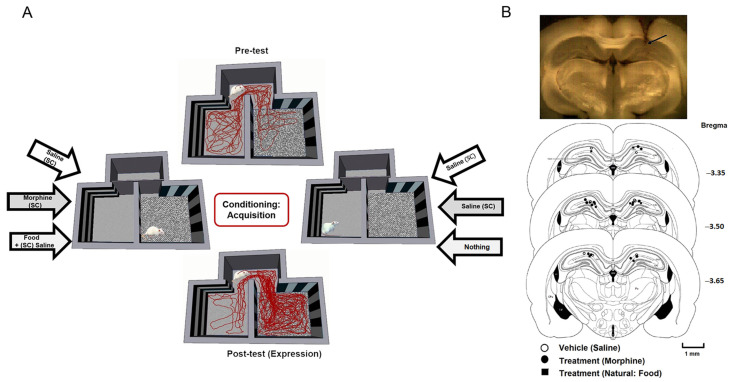
(**A**) Experimental protocols of saline, morphine-CPP, and food-CPP, including the pre-test, acquisition, and post-test. Animal track is marked by red line. (**B**) A coronal photomicrograph of electrode trace showing the CA1 in the rat. Arrow shows the tip of electrode. Three schematic diagrams of the rat brain’s coronal sections, indicating the approximate locations of the electrode sites (○ saline group; ● morphine group; ■ food group). Electrodes were implanted ipsilateral into the right or left side of the CA1 region.

**Figure 2 brainsci-13-00322-f002:**
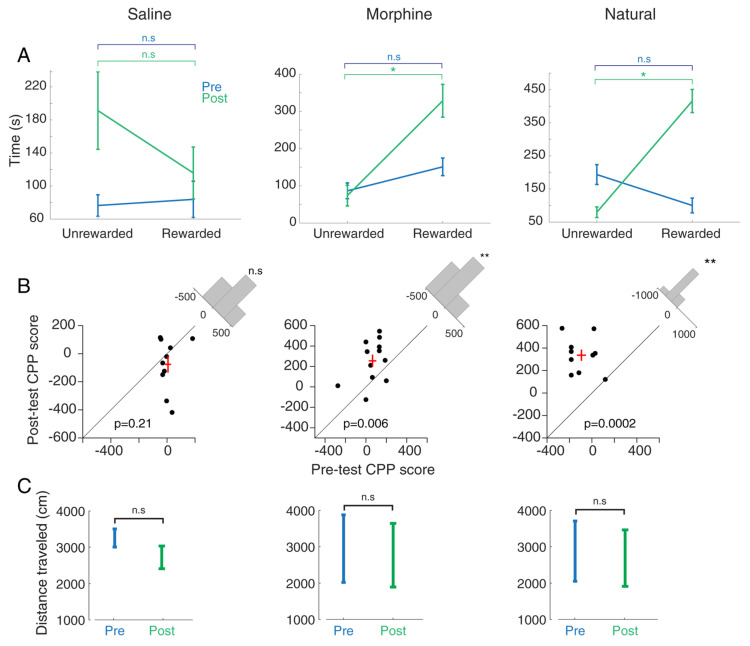
Behavioral data for each saline-CPP, morphine-CPP (as drug), and natural-CPP (food). (**A**) The time spent in rewarded compartment compared to unrewarded compartment during the pre-test (blue color) and post-test (green color). (**B**) Comparing the CPP score of the post-test with the pre-test. Black dots show the numbers of rats in each group; Red cross indicates the average of CPP score (**C**) Comparing the distance traveled between pre-test and post-test. Morphine injection (5 mg/kg, s.c.) induced CPP, and injected rats had a greater tendency to stay in the rewarded (morphine-paired) compartment vs. unrewarded (non-drug) compartment during post-test ((**A**); middle panel). Conditioning by biscuit (natural reward) induced CPP; rats received biscuit during conditioning days had a greater tendency to stay in the rewarded (food-paired) compartment vs. unrewarded (no food) compartment in the post-test session ((**A**); right panel). Rats that received morphine had a higher CPP score in the post-test compared to pre-test ((**B**); middle panel). Rats that received biscuit as a natural reward showed a higher CPP score in the post-test than the pre-test ((**B**); right panel). * *p* < 0.05 as compared with the unrewarded compartment. ** *p* < 0.01 as compared with the pre-test session. n.s. as not significant.

**Figure 3 brainsci-13-00322-f003:**
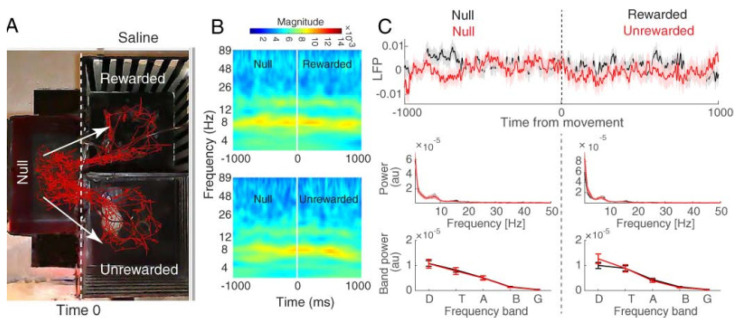
Hippocampal CA1 local field potential of the saline group. (**A**) Behavioral tracking of animals during LFP recording in the post-test session of CPP (red line), animals received saline in both rewarded and unrewarded sides. The dashed line indicates the time that the rat chooses to enter rewarded or unrewarded side from null. The time window was defined as one millisecond before and after the entrance to each side (null–rewarded and null–unrewarded sessions). (**B**) Time–frequency representation based on CWT averaged across all recording trials and aligned to chamber entry. Time zero is defined as the time of entrance to each side from null (shown by the dashed line). (**C**) Total averaged LFP signals (top) and mean power distributions (bottom) for the approach to the rewarded (black) and unrewarded (red) compartment calculated over a time window of one second before and one second after chamber entry. Shaded areas represent SEM.

**Figure 4 brainsci-13-00322-f004:**
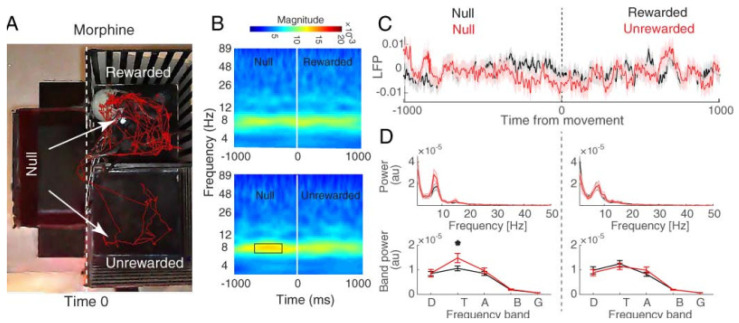
Hippocampal CA1 local field potential of morphine group. (**A**) Behavioral tracking of animals during LFP recording in the post-test session of CPP (red line), animals received morphine in the rewarded compartment while receiving saline in the unrewarded compartment. The dashed line indicates the time that the rat chooses to enter the rewarded or unrewarded side from null. The time window was defined as one millisecond before and after the entrance to each side (null–rewarded and null–unrewarded sessions). (**B**) Time–frequency representation based on CWT averaged across all recording trials and aligned to chamber entry. This figure shows the earliest modulation of theta frequency before time zero (−727 to −153 ms). Power in theta band peaked before the entrance to unrewarded side (top). Time zero is defined as the time of entrance to each side from null (shown by the dashed line). (**C**) Total averaged waveform (top) and (**D**) mean power distributions (bottom) for the approach to the rewarded (black) and unrewarded (red) compartment calculated over a time window of 1 s before and 1 s after chamber entry. Results indicate that mean theta power is increased before the entrance to the unrewarded compartment. Shaded areas represent SEM. * *p* < 0.05 as compared with the null–rewarded session.

**Figure 5 brainsci-13-00322-f005:**
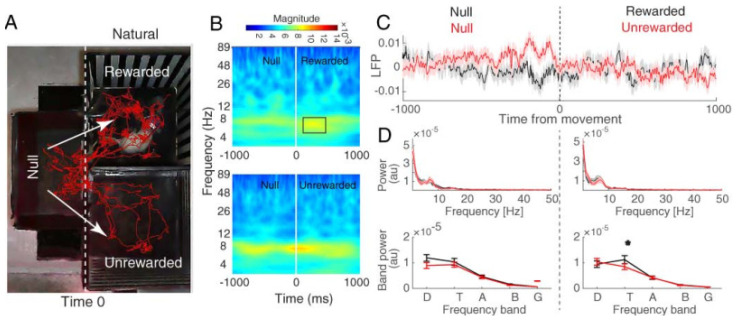
Hippocampal CA1 local field potential of the natural group. (**A**) Behavioral tracking of animals during LFP recording in the post-test session of CPP (red line), animals received a biscuit as a natural reward in the rewarded compartment while receiving saline in the unrewarded compartment. The dashed line stands for when the rat starts to enter into the rewarded or unrewarded side. The time window was defined as one millisecond before and after the entrance to each side (dashed line). (**B**) Time–frequency representation based on CWT averaged across all recording trials and aligned to chamber entry. This figure shows the peak of theta frequency after zero (136–420 ms window). Time zero is defined as the time of entrance to each side from null (shown by the dashed line)**.** (**C**) Total averaged waveform (top) and (**D**) mean power distributions (bottom) for the approach to the rewarded (black) and unrewarded (red) compartment calculated over a time window of 1 s before and after chamber entry. Results indicate that mean theta power is increased after the entrance to the rewarded compartment. Shaded areas represent SEM. * *p* < 0.05 as compared with the unrewarded compartment.

**Figure 6 brainsci-13-00322-f006:**
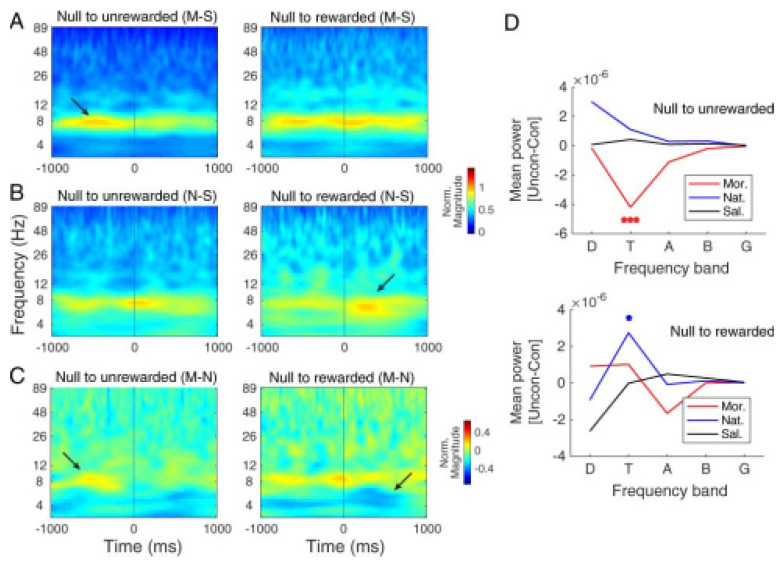
Comparison of hippocampal CA1 local field potential with morphine and natural reward. The left panel shows the time–frequency representation of (**A**) difference between morphine and saline (as control) groups, (**B**) difference between natural (food) and saline (as control) groups, (**C**) and the difference between morphine and natural groups in the time window one second before and after the entrance to rewarded and unrewarded compartments. (**C**) Red indicates greater power of theta frequency before the entrance to the unrewarded compartment (left side), and the blue color indicates greater power theta frequency after the entrance to the rewarded compartment (right side). (**D**) Mean power distributions for saline (black), morphine (red), and natural (blue) reward calculated over a time window of one second before and after unrewarded compartment entry (top) and rewarded compartment entry (bottom). *** *p* < 0.001 as compared with saline and natural. * *p* < 0.05 as compared with saline and morphine.

## Data Availability

Not applicable.

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
