# Peer review of "Selective Modulation of Hippocampal Theta Oscillations in Response to Morphine versus Natural Reward"

_brainsci, 2023, doi:10.3390/brainsci13020322_

Round 1

Reviewer 1 Report

In this work, the authors analyzed changes of hippocampal theta oscillations in rats rewarded with morphine and food. They concluded that drug and food rewards differentially affect the behaviors of tested rats, which is associated with differences in theta dynamic of the hippocampal region. Although I feel that the results are interesting and the manuscript is well written, there are several drawbacks that limit my enthusiasm.

1. In section 2.3.1, it is unclear how rats in the saline group were conditioned. It seems that the authors only described conditioning of morphine and food groups.

2. Importantly, I cannot find the numbers of rats used in each condition and how many independent experiments were performed for different analyses. This information should be provided when reporting statistical data.

3. In Supplementary Figure 1, the authors compared conditioning scores among the 3 groups of rats (saline, morphine, and food). I wonder if the same comparison could be done for other parameters such as times and distance traveled, etc.

4. To support their conclusions, the authors should use pharmacological agents to inhibit hippocampal activity and then test rat behaviors.

5. The authors need to thoroughly check the list of papers they cited. It is not acceptable that many references are incomplete, incorrect, or cannot be found in PubMed with the information provided.

Author Response

In this work, the authors analyzed changes of hippocampal theta oscillations in rats rewarded with morphine and food. They concluded that drug and food rewards differentially affect the behaviors of tested rats, which is associated with differences in theta dynamic of the hippocampal region. Although I feel that the results are interesting and the manuscript is well written, there are several drawbacks that limit my enthusiasm.

We would like to appreciate the time you spent on our manuscript to improve its quality and providing us with your valuable feedback. We would like to mention that we have gone through all the points and made the necessary changes and given our explanations here as well.

  1. In section 2.3.1, it is unclear how rats in the saline group were conditioned. It seems that the authors only described conditioning of morphine and food groups.
  • Many thanks for your comment. Conditioning phase of saline group had been briefly described in the section 2.3.1 “As a control group in the saline group, animals just received saline in either the main compartment”. We also have added more information regarding the conditioning phase of saline group at the end of section 2.3.1 in blue color “Each animal was injected with saline (1 mg/kg, s.c.) in the morning before being placed in one of the main CPP chambers for 45 minutes. Six hours later, it was again injected with saline and confined to another CPP chamber for 45 minutes. The conditioning phase lasted for three days and consisted of two 45-minute sessions per day.” Please kindly let us know if it needs more clarification.
  1. Importantly, I cannot find the numbers of rats used in each condition and how many independent experiments were performed for different analyses. This information should be provided when reporting statistical data.
  • We appreciate your valuable comment. We have used 36 male Wistar rats in the present study; it has added to the section 2.1, 2.5, 3.1 and other reported statistical data. The number of animals in each saline, morphine and food groups was considered 12 per group. During the pre-test phase, the animals that spent more than 70% of the total test time in one of the compartments were excluded from the study for having an initial bias (three in total). And also the electrode tip traces were localized using a light microscope and were confirmed using rat brain atlas. Only the animals with confirmed electrode placements were included in the data analysis (one rat was excluded from the study). Totally four rats were excluded from the study. Please kindly see the section 2.5 including the following sentence regarding the number of animals in each group; “The number of involved animals in data analysis in the saline, morphine and natural groups was 10, 12 and 10, respectively” We have edited the manuscript by mentioning the number of animals throughout the manuscript in blue color.

Thank you again for your constructive comment to improve our manuscript.

  1. In Supplementary Figure 1, the authors compared conditioning scores among the 3 groups of rats (saline, morphine, and food). I wonder if the same comparison could be done for other parameters such as times and distance traveled, etc.
  • Many thanks for your comment. The same comparison had done for distance traveled in the supplementary figure 1. B and described in the caption as follow “(B) Comparing distance traveled among saline, morphine, and natural groups in the post-test phase; this figure indicates that there is no significant difference among these groups.”

Based on your comment we also have done the same comparison for the “time” spent in rewarded compartment during post phase among saline, morphine and food group. Manuscript and supplementary figure.1 has track changed and updated in blue color. Please find the edition in the section 3.1. and supplementary figure 1. Please kindly let us know if you need more information.

  1. To support their conclusions, the authors should use pharmacological agents to inhibit hippocampal activity and then test rat behaviors.
  • We appreciate your valuable comment. It would be very helpful to apply pharmacological agents specially antagonists of reward associated receptors into hippocampus to investigate the rat behavior and theta activity in acquisition and expression phase of morphine- and food- CPP. Our previous studies show that OX2rs in the CA1 region of hippocampus are involved in the development of the acquisition and expression of morphine CPP. Results demonstrate that intra-CA1 administration of the OX2r antagonist attenuates the induction of morphine CPP during the acquisition and expression phases [1]. Also we showed that intra-CA1 administration of the OX1r antagonist attenuates the expression of morphine-induced CPP [2]. We showed that the administration of D1- and/or D2-like dopamine receptor antagonists (SCH23390 and sulpiride, respectively), significantly decreased the acquisition [3], expression [4] of morphine-CPP and Sulpiride shortened the extinction phase of the morphine-induced CPP. These results indicated that the antagonism of these receptors can reduce the rewarding properties of morphine such as acquisition and reinstatement. It would be also very interesting to investigate the role of these receptors such as dopaminergic, glutamatergic, orexinergic and cannabinoid receptors in food-CPP beside the morphine-CPP and it would be more interesting to investigate hippocampal activity during animal’s behavior in a comparative study between morphine-and food-CPP. We would like to mention that your suggestion will be considered as a new research line in our laboratory. Thank you again for your valuable comment.
  1. Sadeghi, B., S. Ezzatpanah, and A. Haghparast, Effects of dorsal hippocampal orexin-2 receptor antagonism on the acquisition, expression, and extinction of morphine-induced place preference in rats. Psychopharmacology, 2016. 233(12): p. 2329-2341.
  2. Farahimanesh, S., S. Karimi, and A. Haghparast, Role of orexin-1 receptors in the dorsal hippocampus (CA1 region) in expression and extinction of the morphine-induced conditioned place preference in the rats. Peptides, 2018. 101: p. 25-31.
  3. Assar, N., et al., D1-and D2-like dopamine receptors in the CA1 region of the hippocampus are involved in the acquisition and reinstatement of morphine-induced conditioned place preference. Behavioural brain research, 2016. 312: p. 394-404.
  4. Nazari-Serenjeh, F., et al., Effects of dopamine D1-and D2-like receptors in the CA1 region of the hippocampus on expression and extinction of morphine-induced conditioned place preference in rats. Behavioural Brain Research, 2021. 397: p. 112924.

  1. The authors need to thoroughly check the list of papers they cited. It is not acceptable that many references are incomplete, incorrect, or cannot be found in PubMed with the information provided.
  • Many thanks for your precise comment. We have checked the references list and have edited and provided the correct references. Please kindly find the edited references in blue color throughout the manuscript.

Thank you again for all your precise, pertinent, and constructive suggestions/comments to improve the entire of our revised manuscript. Hoping that the revised version can meet your criticism, we remain at your disposal for any further eventual modification.

We would appreciate any other modifications and suggestions raised by the referee and editorial board.

Reviewer 2 Report

This study is timely and scientifically sound and properly written, following all the guidelines for publications of scientific articles and it is within the scope of the journal. The abstract of the manuscript fully covers all parts of the manuscript. The figures additionally facilitate the full evaluation of the content contained in the manuscript and significantly increase its value. The introduction provides an interesting admission to the topic. The methods are described in a detailed and clear manner. The research results are presented in a comprehensible and factual way. The discussion contains the most important information necessary to draw conclusions from the conducted research. In my opinion Authors should only:

·       add the information how many animals were kept in one cage as this may affect their behavior in the CPP apparatus

·       add the information how many animals were in each study group

·       add information why 1-way ANOVA was used and not 2-way ANOVA. The Authors should provide the values for obtained ANOVA analysis

·       The Authors should also add a point to the discussion about opioid receptors and their antagonists and their influence on the rewarding effects of morphine or natural rewards. I suggest using and citation the positions:

1.       Gibula-Tarlowska E, Kedzierska E, Piechura K, Silberring J, Kotlinska JH. The influence of a new derivate of kisspeptin-10 - Kissorphin (KSO) on the rewarding effects of morphine in the conditioned place preference (CPP) test in male rats. Behav Brain Res. 2019 Oct 17;372:112043. doi: 10.1016/j.bbr.2019.112043. Epub 2019 Jun 18. PMID: 31226311.

2.       Łupina M, Talarek S, Kotlińska J, Gibuła-Tarłowska E, Listos P, Listos J. The role of linagliptin, a selective dipeptidyl peptidase-4 inhibitor, in the morphine rewarding effects in rats. Neurochem Int. 2020 Feb;133:104616. doi: 10.1016/j.neuint.2019.104616. Epub 2019 Dec 3. PMID: 31809774.

3.       Liang J, Li Y, Ping X, Yu P, Zuo Y, Wu L, Han JS, Cui C. The possible involvement of endogenous ligands for mu-, delta- and kappa-opioid receptors in modulating morphine-induced CPP expression in rats. Peptides. 2006 Dec;27(12):3307-14. doi: 10.1016/j.peptides.2006.08.011. Epub 2006 Nov 9. Erratum in: Peptides. 2007 Mar;28(3):722-3. PMID: 17097192.

·       add the section limitations of the study

Author Response

This study is timely and scientifically sound and properly written, following all the guidelines for publications of scientific articles and it is within the scope of the journal. The abstract of the manuscript fully covers all parts of the manuscript. The figures additionally facilitate the full evaluation of the content contained in the manuscript and significantly increase its value. The introduction provides an interesting admission to the topic. The methods are described in a detailed and clear manner. The research results are presented in a comprehensible and factual way. The discussion contains the most important information necessary to draw conclusions from the conducted research. In my opinion Authors should only:

We would like to appreciate the time you spent on our article to improve the quality of the content and providing us with your valuable feedback. We would like to mention that we have gone through all the points and made the necessary changes and given our explanations here as well.

  1. add the information how many animals were kept in one cage as this may affect their behavior in the CPP apparatus
  • Thank you so much for your interesting comment. As you have mentioned he conditions in which the animal is kept during the experiment are critical, as it affects the animal's behavior. In the present study, two rats were kept in each cage to prevent the stress caused by loneliness. To improve the clarity of this matter, we have mentioned “(two rats per cage)” in the section 2.1 in blue color.

  1. add the information how many animals were in each study group
  • We appreciate your valuable comment. We have used 36 male Wistar rats in the present study; it has added to the section 2.1. The number of animals in each saline, morphine and food groups was considered 12 per group. During the pre-test phase, the animals that spent more than 70% of the total test time in one of the compartments were excluded from the study for having an initial bias (three in total). And also the electrode tip traces were localized using a light microscope and were confirmed using rat brain atlas. Only the animals with confirmed electrode placements were included in the data analysis (one rat was excluded from the study). Totally four rats were excluded from the study. Please kindly see the section 2.5 including the following sentence regarding the number of animals in each group; “The number of involved animals in data analysis in the saline, morphine and natural groups was 10, 12 and 10, respectively” We have edited the manuscript by mentioning the number of animals throughout the manuscript in blue color.

Thank you again for your constructive comment to improve our manuscript

  1. add information why 1-way ANOVA was used and not 2-way ANOVA. The Authors should provide the values for obtained ANOVA analysis
  • Many thanks for your comment. The only independent variable in this study was the type of reward, such as saline, morphine, or food. Therefore, we used one-way ANOVA to examine separately each of the following indices, including CPP score, distance traveled, time, mean power among three groups. We have added the values of ANOVA throughout manuscript in blue color.
  1. The Authors should also add a point to the discussion about opioid receptors and their antagonists and their influence on the rewarding effects of morphine or natural rewards. I suggest using and citation the positions:
  2. Gibula-Tarlowska E, Kedzierska E, Piechura K, Silberring J, Kotlinska JH. The influence of a new derivate of kisspeptin-10 - Kissorphin (KSO) on the rewarding effects of morphine in the conditioned place preference (CPP) test in male rats. Behav Brain Res. 2019 Oct 17;372:112043. doi: 10.1016/j.bbr.2019.112043. Epub 2019 Jun 18. PMID: 31226311.
  3. Łupina M, Talarek S, Kotlińska J, Gibuła-Tarłowska E, Listos P, Listos J. The role of linagliptin, a selective dipeptidyl peptidase-4 inhibitor, in the morphine rewarding effects in rats. Neurochem Int. 2020 Feb;133:104616. doi: 10.1016/j.neuint.2019.104616. Epub 2019 Dec 3. PMID: 31809774.
  4. Liang J, Li Y, Ping X, Yu P, Zuo Y, Wu L, Han JS, Cui C. The possible involvement of endogenous ligands for mu-, delta- and kappa-opioid receptors in modulating morphine-induced CPP expression in rats. Peptides. 2006 Dec;27(12):3307-14. doi: 10.1016/j.peptides.2006.08.011. Epub 2006 Nov 9. Erratum in: Peptides. 2007 Mar;28(3):722-3. PMID: 17097192.
  • Thank you for your constructive comment and sharing valuable studies and information. We have discussed about opioid receptors and their antagonists and their influence on the rewarding effects of morphine or natural rewards in discussion section including the suggested references from your side. Please kindly find the following paragraphs in blue color in the manuscript.

“It has been demonstrated that ICV injections of m-, d-, and k-opioid agonists, or microinjections of these agonists into the NAc shell, prevent morphine-induced CPP in a dose dependent manner [73]. Kisspeptin-10 – kissorphin (KSO) inhibits acquisition, expression of morphine-CPP by its anti-opioid activity [74]. A study indicates that μ-opioid receptor is associated with response to food reward in human [75]. Opioid agonists enhance food intake and hedonic responses to palatable foods, while opioid antagonists de-crease them [76]. It would be interesting to design a comparative study to examine and compare the role of opioid receptors in response to morphine and food reward.

Increasing endogenous glucagon-like peptide-1 (GLP-1) levels in the CNS could attenuate the rewarding effects of both morphine [77] and food [78], likely by activating pre-synaptic GLP-1 receptors on glutamatergic terminals, which facilitate synaptic excitation of dopamine neurons in VTA.”

We appreciate it if you have more suggestion to improve this section.

  1. add the section limitations of the study
  • Thank you for your comment. We designed this study to start a new research line in comparative studies in natural- and drug-induced reward in our laboratory. We believe that it would be more interesting to investigate the neural activity of hippocampal CA1 in the same animal in self-administrative task by both single and LFP recording. Using multi array recording will be suggested to improve the quality of data pull. Optogenetic inhibition of hippocampal CA1 inputs from reward associated regions could reveal which circuit is involved in morphine (as drug) and food (as natural) induced rewards. We had limitation for the afore-mentioned techniques.

Please kindly find this information at the end of discussion in blue color.

Thank you again for all your precise, pertinent, and constructive suggestions/comments to improve the entire of our revised manuscript. Hoping that the revised version can meet your criticism, we remain at your disposal for any further eventual modification.

We would appreciate any other modifications and suggestions raised by the referee and editorial board.

Round 2

Reviewer 1 Report

Authors have addressed the issues raised in my previous review, although I am not sure they followed the reference style of the journal.

Author Response

Dear Professor Stephen D. Meriney,

Thank you for constructive comments about the manuscript. We have modified the manuscript accordingly and have added all the necessary explanations to the manuscript based on the points raised by the reviewers, which are all now marked in blue color. We have added more references based on the academic editor’s and reviewers’ comments as well. We would appreciate any other modifications and suggestions raised by referee and editorial board.

Response to reviewer  #1 Comments

Authors have addressed the issues raised in my previous review, although I am not sure they followed the reference style of the journal.

  • Thank you for your comment. We have changed and edited references based on the reference style of brain sciences.     

We would appreciate any other modifications and suggestions raised by the referee and editorial board.
